# Decoupling between Translational Diffusion and Viscoelasticity in Transient Networks with Controlled Network Connectivity

**DOI:** 10.3390/gels8120830

**Published:** 2022-12-16

**Authors:** Takuya Katashima, Ryunosuke Kobayashi, Shohei Ishikawa, Mitsuru Naito, Kanjiro Miyata, Ung-il Chung, Takamasa Sakai

**Affiliations:** 1Department of Bioengineering, School of Engineering, The University of Tokyo, 7-3-1 Hongo, Bunkyo-ku, Tokyo 113-8656, Japan; 2Department of Materials Engineering, Graduate School of Engineering, The University of Tokyo, 7-3-1 Hongo, Bunkyo-ku, Tokyo 113-8656, Japan; 3Center for Disease Biology and Integrative Medicine, Graduate School of Medicine, The University of Tokyo, 7-3-1 Hongo, Bunkyo-ku, Tokyo 113-8656, Japan; 4Department of Chemistry and Biotechnology, Graduate School of Engineering, The University of Tokyo, 7-3-1 Hongo, Bunkyo-ku, Tokyo 113-8656, Japan

**Keywords:** transient network, viscoelasticity, diffusion, tetra PEG, phenyl boronic acid, diol

## Abstract

The mobility of sustained molecules is influenced by viscoelasticity, which is strongly correlated with the diffusional property in polymeric liquid. However, the study of transient networks formed by a reversible crosslink, which is the viscoelastic liquid, was insufficient due to the absence of a model system. We compare the viscoelastic and diffusional properties of the transient networks, using the model system with controlled network connectivity (Tetra-PEG slime). According to independent measurements of viscoelasticity and diffusion, the root-mean-square distance the polymer diffuses during the viscoelastic relaxation time shows a large deviation from the self-size of the polymer, which is contrary to the conventional understanding. This decoupling between viscoelasticity and diffusion is unique for transient networks, suggesting that the viscoelastic relaxation is not induced by the diffusion of one prepolymer, particularly in the network with low connectivity. These findings will provide a definite basis for discussion to understand the viscoelasticity in transient networks.

## 1. Introduction

Transient networks are three-dimensional networks formed by reversible crosslinks with a finite lifetime, including coordinate bonds, hydrogen bonds, dynamic covalent bonds, etc. Because the reversible bond nature differs from the covalent bond, these crosslinks continuously bind and dissociate, where the translational diffusion of the constitutive polymers is permitted. As a result, the transient networks are not solid, but are liquid showing temporal elasticity (viscoelastic liquid) [1,2,3,4,5,6,7,8]. Because they can slowly dissolve into a solvent and release the entrapped molecules, there have been many applications for drug-delivery carriers [9,10,11,12,13,14,15]. To control the release rate of entrapped molecules, the molecular mechanism of viscoelasticity should be elucidated. This is because matrix viscoelasticity defines the diffusion of entrapped molecules, as described by some fluidics models including the generalized Stokes-Einstein equation [16,17,18,19,20].

In general, the elasticity of the polymeric liquid originates from the orientational anisotropy in polymer chains with entropic elasticity between the ends [21,22,23]. Thus, the shear stress expression of polymeric liquid is as follows:(1)$$\begin{array}{c} \sigma \left(t\right)=3\nu k_{B}T\overset{N}{\underset{n=1}{\sum }}S\left(n,t\right) \end{array}$$

Here, *ν*, *k*_B_, and *T* are the number density of the polymer chains composed of *N* submolecules, the Boltzmann constant, and the absolute temperature, respectively. *S*(*n*,*t*) is called the shear orientational function, defined as:(2)$$\begin{array}{c} S\left(n,t\right)=\langle \frac{1}{a^{2}}u_{x}\left(n,t\right)u_{y}\left(n,t\right)\rangle \end{array}$$
*a* is the average end-to-end distance of a chain. *u_i_*(*n*, *t*) is the end-to-end vector of the *n*-th submolecules in the *i*-th chain at a particular time, *t*. <...> represents the ensemble average for all the chains. Equation (1) means that entropic tension along each chain always occurs. Still, if the spatial distribution of the chains is isotropic, the overall tension is balanced, and no macroscopic stresses can be observed. When the strain is applied, and the chain distribution becomes anisotropic, the balance is broken, resulting in macroscopic stress. On the other hand, the anisotropic distribution disappears through each chain diffusion, which is the origin of the viscoelastic relaxation in polymeric liquid. In general cases, including the unentangled and entangled polymers, the time required for the clearance of the anisotropic distribution (*τ*) is described as:(3)$$\begin{array}{c} \tau \approx \frac{R^{2}}{D_{\mathrm{rotational}}} \end{array}$$

Here, *R* is approximated to be the self-size of the polymer chains (the average end-to-end distance and the contour length of a chain), and *D*_rotaional_ means the rotational diffusion coefficient. Flexible polymers have random coil conformation in a good solvent and are approximated to be a sphere. Under sphere approximation, the “translational” diffusion coefficient, *D*_translational_, is proportional to *D*_rotational_, which indicates that the time that the polymers translationally diffuse about “self-size” agrees with the viscoelastic relaxation time. Many experimental studies have verified this concept in polymer melts and solutions [22,23,24,25,26].

However, there have been few studies on transient networks [3,27]. In a few reports, the root-mean-square distance that constitutive molecules diffuse during the viscoelastic relaxation time was approx. 1µm, much larger than the self-size, which is puzzling from the conventional understanding of polymeric liquid. However, in conventional systems using the telechelic associative polymer and threadlike micelles, the structural parameters including network connectivity, functionality, and network strand length were not defined. As a result, the discussion on the deviation from the conventional understanding was ambiguous [4,28].

Recently, we designed the model transient networks (Tetra-PEG slime) by employing dynamic covalent bonds between phenylboronic acid (FPBA) and diol (GDL) as crosslinks between four-armed polymers [29]. Tetra-PEG slime is composed of Tetra-functional precursors with a narrow distribution, leading to the formation of regular structures with uniform network strand length and functionality. In addition, using symmetric precursors with the same mobility reduces the heterogeneous dynamics compared with the conventional system.

In this study, we report the relationships between linear viscoelasticity and translational diffusion using the Tetra-PEG slime. To estimate the translational diffusion coefficient, we modified small amounts of the end group in Tetra-armed polymers with a fluorescent dye and performed the fluorescent recovery after the photobleaching (FRAP). In a FRAP experiment, fluorescence-labeled molecules are rapidly and permanently bleached in a circular or rectangular region of interest (ROI) using a powerful laser beam. After photobleaching, the bleached and non-bleached molecules are exchanged by diffusion, and fluorescence at ROI gradually recovers. The diffusion coefficients and local mobile/immobile molecule fraction can be estimated using appropriate exponential-type equations. Other methods for estimating translational diffusion have been proposed: pulsed-field gradient nuclear magnetic resonance (PFG-NMR), single-particle tracking (SPT), and so on. PFG-NMR and SPT are easy to measure because further modification with any fluorescence is not required. However, the estimated diffusion coefficient regions are limited in the measurements. Thus, we utilized the FRAP technique in this study.

Here, to discuss the effect of the network connectivity independent of other structural parameters (polymer concentration, network strand length, etc.), we mixed the two precursors in off-stoichiometric ratios [30]. Our findings will help better understand the molecular dynamics dominating the viscoelasticity in the transient networks.

## 2. Results and Discussion

### 2.1. Effects of Network Connectivity on Viscoelasticity

Figure 1 shows the results of dynamic viscoelastic measurements for the Tetra-PEG slimes with various mixing fractions at 30 °C. *G*′ showed the plateau region and the power law *G*′~*ω*^2^, whereas *G*″ demonstrated the symmetric power law behaviors *G*″~*ω* and *G*″~*ω*^−1^ at high and low frequencies, respectively. These characteristics agree well with the prediction of the Maxwellian model, which suggests that viscoelastic relaxation occurs via a unique process. It should be noted that the upturn at high frequencies was observed in *s* = 0.20, which is attributed to the Rouse mode of the dangling chains. As the values of *s* decreased from 0.50, the plateau level of *G*′ decreased, and the peak of *G*″ shifted to high frequencies. In Appendix A, the viscoelasticity of the Alexa-labeled and neat samples overlapped each other, indicating that the effect of the fluorescence modification on the rheological property is negligible.

The acceleration of the viscoelastic relaxation is depicted against the network connectivity in Figure 2. Here, the connectivity is defined as the fraction of the connected end groups against the total end groups at the equilibrium state. According to the kinetic equation of the one-to-one binding reaction, the connectivity is described as:(4)$$\begin{array}{c} p=\left\{1+\frac{1}{C_{\mathrm{end}}\;K}\right\}-{\left[{\left\{1+\frac{1}{C_{\mathrm{end}}\;K}\right\}}^{2}-4s\left(1-s\right)\right]}^{\frac{1}{2}} \end{array}$$

*C*_end_ and *K* are the total end group concentration and the equilibrium constant, respectively. *K* was previously estimated to be 6.4 × 10^2^ M^−1^ by the surface plasmon resonance for the mono-functional PEG-FPBA and PEG-GDL [29,30]. It should be noted that *p* estimated by Equation (4) is just the reaction efficiency of the end groups, which includes the bridging chains and closed loop structures, especially low-connectivity regions.

*τ*_visco_ monotonically decreased with decreasing *p* and became shorter than the dissociation time of the dynamic covalent bond (dashed line), estimated by the surface plasmon resonance [29,30]. The *p*-dependence of the viscoelastic relaxation time is a surprising result contrary to the theoretical predictions by Green, Tobolsky [31], and Yamamoto [32], where the viscoelastic relaxation agrees with the dissociation of the bonds. This is because they assumed that the Rouse dynamics of the network strand are much faster than the bond lifetime, and the orientational anisotropy in the whole system disappears immediately after the dissociation of bonds. This deviation from the theoretical prediction suggests that the viscoelastic relaxation is determined not by the single chain dynamics but by the collective “network” dynamics.

### 2.2. Effects of Network Connectivity on Diffusivity

Figure 3 shows the time development of the fluorescent intensity for the Tetra-PEG slimes in a FRAP measurement. By the laser irradiation at *t* = 0 s, the intensity was reduced stepwise. After the irradiation, the intensity recovered with time and reached the plateau value at approximately 400 s. Here, the fluorescent intensity (*I*) was normalized by the background and bleaching intensities. The recovery curve is fitted by the Kohlrausch-Williams-Watts (KWW) type equation as:(5)$$\begin{array}{c} I=A-B\exp\left\{-{\left(\frac{t}{\gamma \tau_{\mathrm{KWW}}}\right)}^{\beta }\right\} \end{array}$$

Here, *A* is the intensity at the plateau region in the long-time limit, *B* is the reduction of the intensity by photobleaching, *τ*_KWW_ is the characteristic recovery time, *β* is the polydispersity index for the recovery time, and *γ* is the modification factor (=1.5). Generally, the fluorescence photobleaching recovery process for single-component diffusing species is described by a single exponential function for a two-dimensional diffusion. However, the bleaching laser intensity profile is not homogeneous, characterized by a Gaussian function. The exponent of *β* includes this experimental non-ideality of laser intensity (≈0.6). As a result, the apparent relaxation time in the KWW function is accelerated from the accurate diffusion time, which is modified by the constant *γ*. The details are previously described in Appendix [3]. Notably, fitting by Equation (5) the immobilized fraction (*A* in Equation (5)) is estimated to be approx. 0.25–0.3, which roughly agrees with the percolation threshold of the diamond lattice in three-dimensional space. This agreement suggests that the contribution of the percolation networks on diffusion is negligible.

Figure 4 shows the estimated *τ*_KWW_ against *d*^2^. The linear relationship between *τ*_KWW_ and *d*^2^ was observed, indicating that the fluorescence recovery is primarily attributed to the Fickian diffusion of molecules, and the chemical interconversion is negligible. It should be noted that the data in the smallest area in *s* = 0.5 deviated from the linear relationship due to the non-Fickian (anomalous) diffusion in transient networks [5,33,34]. In the discussion below, we utilized the data showing the linear relationship to estimate the translational diffusion coefficient.

For the KWW equation, the analytical solution of the second-order average relaxation time was reported [35] as:(6)〈τ〉2=Γ(2β)Γ(1β)τKWW

The diffusion coefficient, *D*, is estimated using the bleaching area, *d*^2^, as:(7)$$\begin{array}{c} D=\frac{d^{2}}{4{<\tau >}^{2}} \end{array}$$

Figure 5a shows *D* as a function of *S*. At the large *d*^2^ region, *D* is almost independent of *d*^2^ and constant values, reflecting the translational diffusion in two-dimensional space. In the region where <*τ*> was independent of *d*^2^, the translational diffusion coefficient (*D*_translational_) is defined (represented by dashed lines). The estimated *D*_translational_ is plotted against *p* in Figure 5b. *D*_translational_ decreased with increasing *p* in the semi-logarithmic plot, suggesting the collision between stickers restricts the polymer dynamics. This tendency was observed in the previous reports [3,5,33,34], and is quantitatively consistent with the Sticky Rouse model [36].

### 2.3. Comparison between Viscoelasticity and Diffusivity

We independently evaluated the viscoelastic relaxation time and translational diffusion coefficient as a function of network connectivity. To compare both properties directly, we discuss the root-mean-square distance the prepolymers diffuse during the viscoelastic relaxation time (*RMSD*). Assuming the random walk, *RMSD* is described as:(8)$$\begin{array}{c} RMSD={\left\{D_{\mathrm{translational}}\;\tau_{\mathrm{visco}}\right\}}^{1/2} \end{array}$$

Figure 6 shows the *p*-dependence of *RMSD*. In the figure, the dashed line represents the gyration radius of a Tetra-PEG (≈*aN*^ν^; *a* is the Kuhn length of polyethylene glycol (≈0.65 nm [37,38,39,40]), *N* is the Kuhn segment number, *ν* is the excluded volume exponent in good solvent (≈0.57)). *RMSD* is an order of 10^−6^ m and approx. 100 times larger than the gyration radius even in the Tetra-PEG slime with the controlled network structures. This significant difference indicates that the viscoelastic relaxation does not proceed through the diffusive motion of each prepolymer. In this experimental system, the equilibrium connectivity is up to 0.6. Therefore, a mobile sol component in which not all arms are connected to the percolated network exists. It is suggested that the FRAP measurement preferentially detects this mobile component.

On the other hand, viscoelasticity reflects the orientational anisotropy dynamics of the percolated network. For instance, when one prepolymer in a percolated network breaks off and diffuses but is replaced by another prepolymer, the overall orientational anisotropy remains. As a result, a decoupling of diffusion and viscoelasticity occurs. In the figure, *RMSD* approached the gyration radius as *p* increased and will coincide with it in the high *p* limit, which also supports the above discussion.

Notably, in this study, we utilized the Alexa-labeled polymers with an effective functionality of 3 instead of 4. As for diffusion, the effect of decreasing functionality is not negligible. However, according to previous reports, the diffusion coefficient of the tri-functional polymer in transient networks is 2–4 times larger than that of the tetra-functional one [33]. The difference cannot explain the deviation of RMSD from the gyration radius.

## 3. Conclusions

We independently estimated the viscoelasticity and diffusional properties of the transient networks with defined network connectivity (Tetra-PEG slime). Our key findings are as follows: (1) the viscoelastic relaxation time decreased with the network connectivity decreasing (*p*); (2) the translational diffusion coefficient decreased with increasing *p*; (3) the root-mean-square distance the prepolymers diffuse during the viscoelastic relaxation time is 100 times larger than the self-size of the prepolymer, which is contrary to the conventional understanding in polymeric liquid. This decoupling between viscoelasticity and diffusion in transient networks becomes pronounced decreasing the network connectivity, suggesting that the small fraction of the percolated networks is almost immobilized and supports the viscoelasticity of transient networks, where the polymer diffusion does not relax the orientational anisotropy and mainly proceeds through the recombination. On the other hand, decoupling is suppressed at higher connectivity, suggesting that the cooperativity of the networks is enhanced at high connectivity limits. These findings will provide a solid foundation for discussion to understand the molecular picture of viscoelasticity in transient networks from the viewpoint of molecular diffusion.

## 4. Materials and Methods

### 4.1. Synthesis of Fluorescence-Labeled Tetra-Armed Prepolymers

Amine-terminated tetra-armed polyethylene glycol (tetra-PEG-NH_2_) (*M*_w_ = 2.0 × 10^4^ g mol^−1^) was purchased from SINOPEG BIOTECH Co., Ltd. (Xiamen, China), and its end groups were modified with of 4-carboxy-3-fluorophenylboronic acid (FPBA). The details were reported previously [29]. On the other hand, we synthesized the Tetra-PEG-D-(+)-glucono-1,5-lactone (GDL) partially modified with Alexa Fluor 594 (Alexa). The Tetra-PEG-NH_2_ was dissolved in methanol at a concentration of 50 g L^−1^. Separately, Alexa Fluor™ 594 NHS ester (succinimidyl ester) (Alexa-NHS), purchased from Thermo Fisher SCIENTIFIC (Tokyo, Japan), was dissolved in super-dehydrated dimethyl sulfoxide at the concentration of 1 mg/mL. Compared to the molar amount of the amine end group, we added 0.001 times the amount of Alexa-NHS and stirred the mixture overnight at room temperature. Compared to the molar amount of the amine end group, we added 10 times the amount of D-(+)-glucono-1,5-lactone (GDL) and 20 times the amount of triethylamine and stirred the mixture for three days at 35 °C. The resultant solution was poured into a dialysis membrane (molecular weight cut-off: 3500 for 10,000 g mol^−1^) and dialyzed for two days with methanol and water. After passing through a syringe filter (0.45 μm), the solutions were collected and freeze-dried. The synthesis scheme is shown in Figure 7. ^1^H-NMR was used to confirm that the synthesis was complete. The fluorometer confirmed that the modification degree of the Alexa was 0.0444% (shown in Appendix A).

### 4.2. Sample Preparation

Tetra-PEG slimes were prepared by mixing the FPBA- and GDL-modified Tetra-PEG, which were dissolved in phosphate buffer (pH 7.4, 200 mM). Prepolymer concentrations were set to 60 g L^−1^. The two polymer solutions were mixed at various stoichiometrically imbalanced fractions of *s* = 0.20, 0.30, 0.40, 0.50, where *s* = [Tetra-PEG-GDL]/([Tetra-PEG-GDL] + [Tetra-PEG-FPBA). Each reaction was allowed to proceed for 12 h at 25 °C.

### 4.3. Dynamic Viscoelastic Measurements

For the dynamic viscoelastic measurements, Tetra-PEG slime samples were placed on the measuring plate of a stress-controlled rheometer (MCR301; Anton Paar, Graz, Austria) utilizing a cone plate fixture with a 25 mm diameter and a 4° cone angle. The angular frequency dependence (0.1–100 rad s^−1^) of the storage (*G*′) and loss (*G*″) moduli was measured at 30 °C. Before the measurements, the oscillatory shear strain amplitudes were confirmed to be within the linear viscoelasticity range.

### 4.4. Fluorescence Recovery after Photobleaching (FRAP) Measurements

The FRAP technique measured the molecular diffusivity with a laser scanning confocal microscope (Axio Observer (LSM800); ZEISS, Oberkochen, Germany) using the 488 nm laser line at 30 °C. Before the bleaching process, the fluorescence intensity of the target area was measured at low laser intensity to obtain the reference image. Afterward, a circular area with a diameter (*d*) of 20–80 µm was bleached by scanning the laser at its full power 1–15 times. The duration of the bleaching was set to 0.12–3.8 s. The recovery process was recorded by scanning at low laser intensity.

## Figures and Tables

**Figure 1 gels-08-00830-f001:**
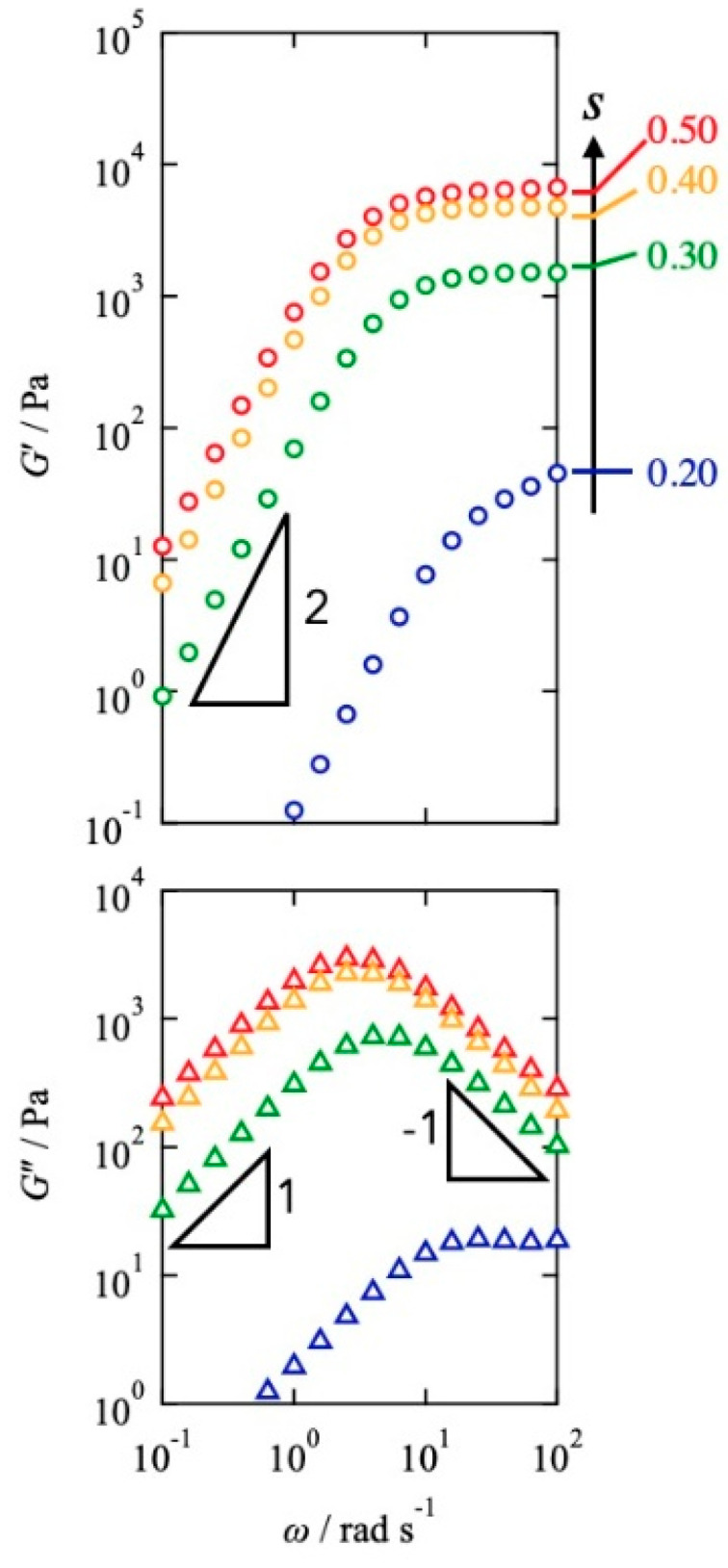
Angular frequency dependence of *G*′ (**top** panel) and *G*″ (**bottom** panel) for the Tetra-PEG slimes with various *s* at 30 °C.

**Figure 2 gels-08-00830-f002:**
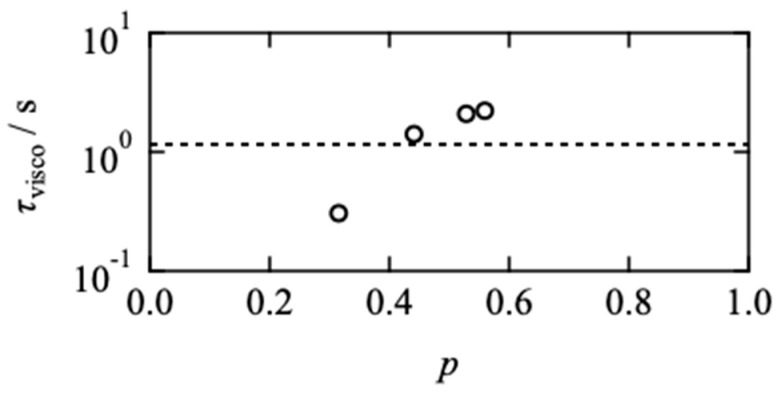
Viscoelastic relaxation time (*τ*_visco_) as a function of *p*. Dashed line represents the dissociation time of end groups estimated by surface plasmon resonance in our previous report [29,30].

**Figure 3 gels-08-00830-f003:**
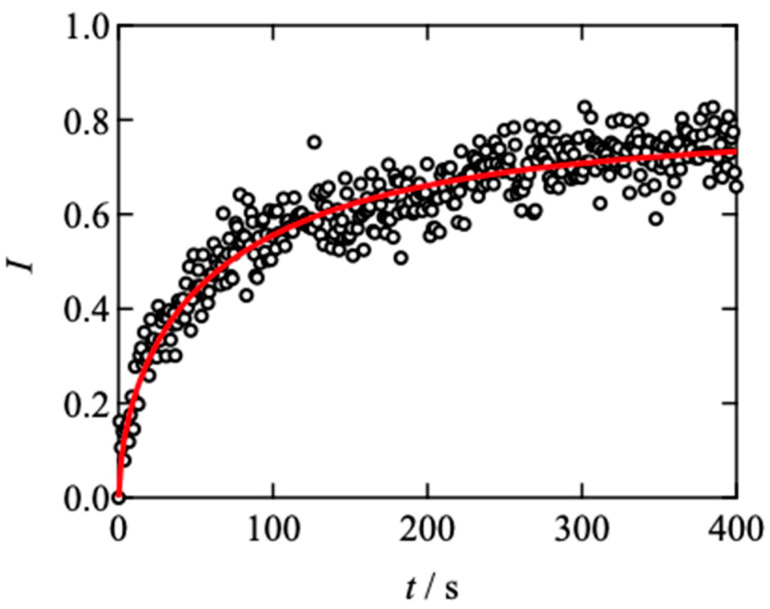
Representative results of fluorescence recovery after photobleaching for Tetra-PEG slime (*s* = 0.5, *d* = 2.5 µm) at 30 °C. The dashed line represents the fitting results using Equation (5).

**Figure 4 gels-08-00830-f004:**
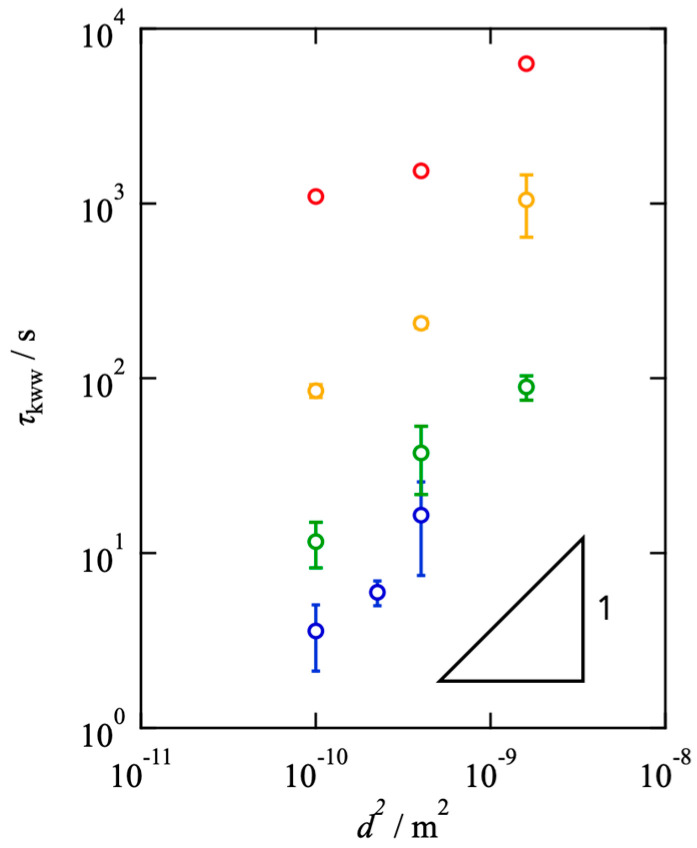
Estimated *τ*_KWW_ as a function of *d^2^* for the Tetra-PEG slimes with various *s* (red: *s* = 0.5, yellow: *s* = 0.4, green: *s* = 0.3, blue: *s* = 0.2).

**Figure 5 gels-08-00830-f005:**
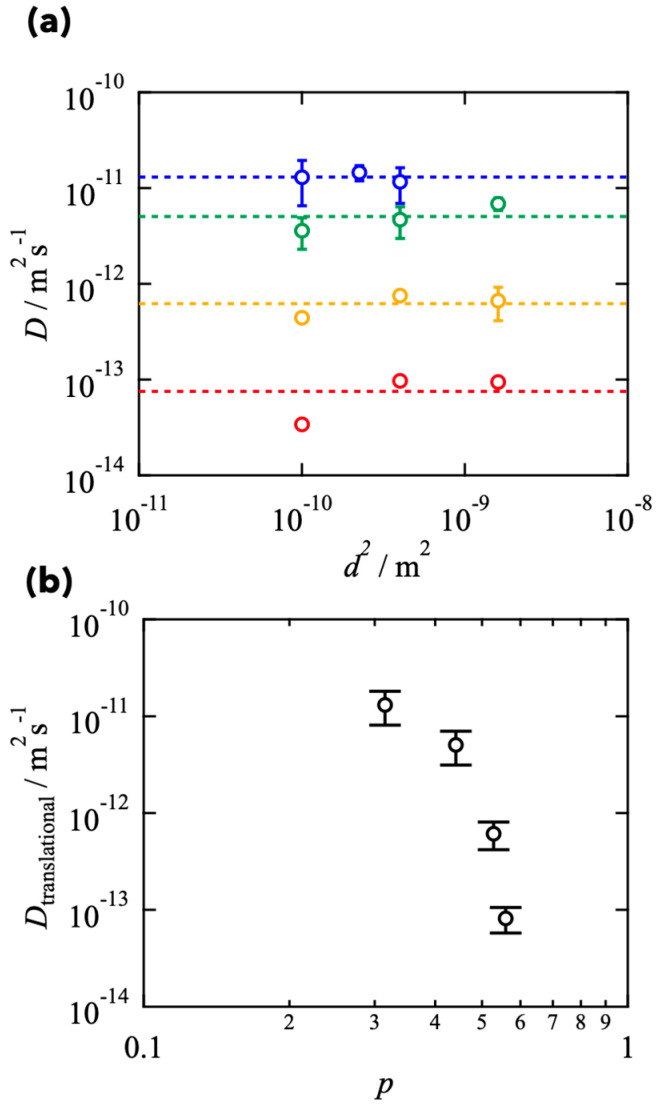
(**a**) *d*^2^-dependence of diffusion coefficient (*D*) for the Tetra-PEG slimes with various *s* (red: *s* = 0.5, yellow: *s* = 0.4, green: *s* = 0.3, blue: *s* = 0.2). (**b**) Estimated translational diffusion coefficient (*D*_translational_) as a function of *p*.

**Figure 6 gels-08-00830-f006:**
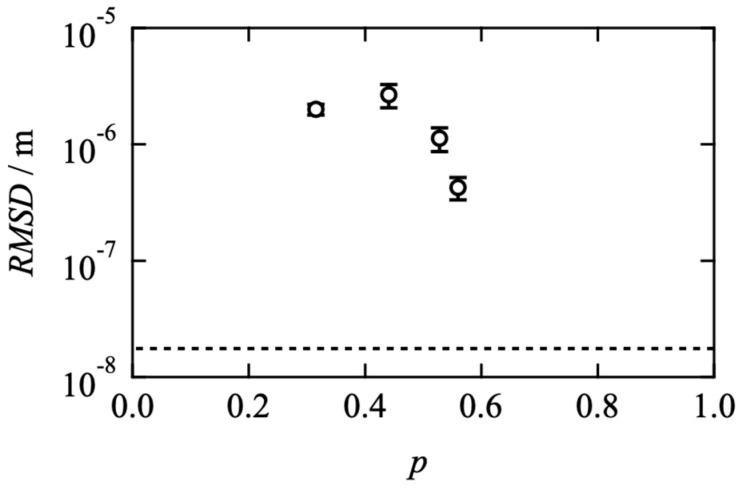
Root-mean-square distance the prepolymers diffuse during the viscoelastic relaxation time (*RMSD*) as a function of *p*. The dashed line represents the gyration radius of a four-armed precursor chain.

**Figure 7 gels-08-00830-f007:**
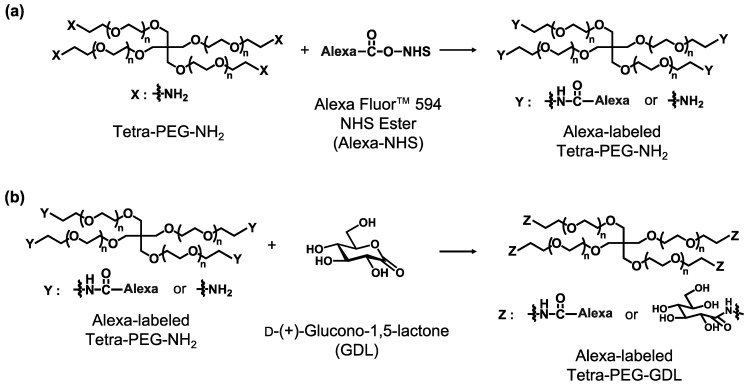
Reaction Scheme of Tetra-PEG-NH_2_ and Gluconolactone (**a**,**b**).

## Data Availability

Not applicable.

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
