# Peer review of "Decoupling between Translational Diffusion and Viscoelasticity in Transient Networks with Controlled Network Connectivity"

_gels, 2022, doi:10.3390/gels8120830_

Round 1
Reviewer 1 Report
The manuscript is well writen and presented. Any big mistakes or flaws were found. I just have to remember that the whole text should be written using the same font.
Author Response
To Reviewer 1
First, we would like to thank the reviewer for the helpful comments. We have addressed the concerns in our revised manuscript. The revision in the manuscript was colored yellow. The detailed responses are given below:
Comments:
The manuscript is well written and presented. Any big mistakes or flaws were found. I just have to remember that the whole text should be written using the same font.
Answer:
Thank you for the positive comment. We have revised the font in the manuscript.
*** End of response to comments from Reviewer 1 ***
Reviewer 2 Report
This is an interesting piece of work on the dynamics of Transient Networks. The authors choose model systems based on tetra-functional (four arms) polymers able to connect at the arms end by dynamic covalent bonds and form a transient Network.
The manuscript is well written and can be published after minor revision. There are few points on which the authors should elaborate before publication:
1) As far as I understood, the authors consider that all connected arms (defined as connectivity, equation 4) contribute to the viscoelastic behavior of the system. This assumption neglects the formation of closed loops between two neighboring tetra-functional PEG polymers. In fact, when a Tetra-PEG-GDL polymer forms a GDL-FPBA bond with a neighboring Tetra-PEG-FPBA polymer, there is a high probability to form a second GDL-FPBA bond between the same two polymers. This reduces the overall contribution of GDL-FPBA bonds to the network viscoelasticity.
2) The FRAP experiments follow the behavior of Alexa-labeled polymers. However, these labelled polymers have an effective functionality of 3 instead of 4 for the unlabeled molecules. How does this difference in functionality influence the relaxation time measured by FRAP and the comparison with the rheology relaxation time?
3) Line 38: “inclusive molecules”. Do the authors mean “entrapped molecules”
4) Line 225: what does “prepolymer” mean? Is it the non-functionalized Tetra-PEG?
Author Response
To Reviewer 2
First, we would like to thank the reviewer for the helpful comments. We have addressed the concerns in our revised manuscript. The revision in the manuscript was highlighted in blue. The detailed responses are given below:
Reviewer’s comment:
This is an interesting piece of work on the dynamics of Transient Networks. The authors choose model systems based on tetra-functional (four arms) polymers able to connect at the arms end by dynamic covalent bonds and form a transient Network.
The manuscript is well written and can be published after minor revision. There are few points on which the authors should elaborate before publication:
C1. As far as I understood, the authors consider that all connected arms (defined as connectivity, equation 4) contribute to the viscoelastic behavior of the system. This assumption neglects the formation of closed loops between two neighboring tetra-functional PEG polymers. In fact, when a Tetra-PEG-GDL polymer forms a GDL-FPBA bond with a neighboring Tetra-PEG-FPBA polymer, there is a high probability to form a second GDL-FPBA bond between the same two polymers. This reduces the overall contribution of GDL-FPBA bonds to the network viscoelasticity.
A1. We appreciate the reviewer for the useful comments. We agree with the reviewer that closed loops are formed. especially in low-connectivity regions. We realized that our description was insufficient. We have added the possible contribution of closed loops in the revised manuscript.
C2. The FRAP experiments follow the behavior of Alexa-labeled polymers. However, these labelled polymers have an effective functionality of 3 instead of 4 for the unlabeled molecules. How does this difference in functionality influence the relaxation time measured by FRAP and the comparison with the rheology relaxation time?
A2. We appreciate the reviewer’s pointing out. As for rheology, we confirmed that the effect of functionality is negligible shown in Figure S1, which is due to the low modification rate. On the other hand, as for diffusion, the effect is not negligible. However, according to previous reports, the diffusion coefficient of the tri-functional polymer in associative networks is larger than that of the tetra-functional one. However, the ratio was 2-4 times, less than 10 times. The difference cannot explain the decoupling phenomenon of this study. We realized that the effect of functionality is insufficient. We have added the description in the revised manuscript.
C3. Line 38: “inclusive molecules”. Do the authors mean “entrapped molecules”
A3. Thank the reviewer for the comment. We meant the “inclusive molecules” as “ entrapped molecules.” We have revised the phrase in the main text.
C4. Line 225: what does “prepolymer” mean? Is it the non-functionalized Tetra-PEG?
A4. We appreciate the reviewer’s pointing out. We meant the term prepolymer as Tetra-PEG. We have revised the phrase to avoid confusion.
*** End of response to comments from Reviewer 2 ***
Reviewer 3 Report
References are usually placed at the end of the referenced text, and before the period, for example, in lines 36, 38, 41.
Correct font and font size on lines 99 - 101.
The Celcius degree symbol should be separated from the numerical value; lines 16, 105, 119, 125, 130, 139, etc.
Conclusions are weak, and they should be improved.
Author Response
To Reviewer 3
First, we would like to thank the reviewer for the helpful comments. We have addressed the concerns in our revised manuscript. The revision in the manuscript was highlighted in green. The detailed responses are given below:
C1. References are usually placed at the end of the referenced text, and before the period, for example, in lines 36, 38, 41.
Correct font and font size on lines 99 - 101.
The Celcius degree symbol should be separated from the numerical value; lines 16, 105, 119, 125, 130, 139, etc.
A1. Thank you for the detailed comments. We have revised the corresponding parts in the manuscript.
C2. Conclusions are weak, and they should be improved.
A2. We thank the reviewer’s suggestion. We realized that the conclusion is weak, and have revised the conclusion.
*** End of response to comments from Reviewer 3 ***
Reviewer 4 Report
The topic of paper is very interesting and the paper is also very well written however there are some suggestion that might improve the paper
[1] introduction needs more discussion on FRAP topics such as its differences and limitation against other methods; following references can be used for this aim: Details of modeling of FRAP can als be found in the following references.
*Sprague, B. L., & McNally, J. G. (2005). FRAP analysis of binding: proper and fitting. Trends in cell biology, 15(2), 84-91.
* Moud, A. A. (2022). Fluorescence Recovery after Photobleaching in Colloidal Science: Introduction and Application. ACS Biomaterials Science & Engineering, 8(3), 1028-1048.
[2] In section 2.1; please fix the font size for sentence"Alexa 99 Fluor™ 594 NHS ester (succinimidyl ester) (Alexa-NHS) was dis- 100 solved in super-dehydrated dimethyl sulfoxide at the concentration 101 of 1 mg/mL. C"
[3] the sentences "These findings will pro- 254 vide a solid foundation for discussion to understand the molecular picture of viscoelastic- 255 ity in transient networks from the viewpoint of molecular diffusion" is very interesting; authors can also estimate immobile particle percentage as well using models that can be found in the reference mentioned earlier.
Author Response
To Reviewer 4
First, we would like to thank the reviewer for the helpful comments. We have addressed the concerns in our revised manuscript. The revision in the manuscript was highlighted in purple. The detailed responses are given below:
Reviewer’s comment:
The topic of paper is very interesting and the paper is also very well written however there are some suggestion that might improve the paper.
C1. introduction needs more discussion on FRAP topics such as its differences and limitation against other methods; following references can be used for this aim: Details of modeling of FRAP can also be found in the following references.
*Sprague, B. L., & McNally, J. G. (2005). FRAP analysis of binding: proper and fitting. Trends in cell biology, 15(2), 84-91.
* Moud, A. A. (2022). Fluorescence Recovery after Photobleaching in Colloidal Science: Introduction and Application. ACS Biomaterials Science & Engineering, 8(3), 1028-1048.
A1. Thank you for your kind suggestion. We realized that the introduction of FRAP was insufficient. We have added the discussion on FRAP in the revised manuscript.
C2. In section 2.1; please fix the font size for sentence"Alexa 99 Fluor™ 594 NHS ester (succinimidyl ester) (Alexa-NHS) was dis- 100 solved in super-dehydrated dimethyl sulfoxide at the concentration 101 of 1 mg/mL.
A2. We have revised the font sizes.
C3. the sentences "These findings will provide a solid foundation for discussion to understand the molecular picture of viscoelasticity in transient networks from the viewpoint of molecular diffusion" is very interesting; authors can also estimate immobile particle percentage as well using models that can be found in the reference mentioned earlier.
A3. We thank you for your kind suggestion. We estimated the immobilized fraction and added the discussion in the revised manuscript.
*** End of response to comments from Reviewer 4 ***